# [Re] Distilling Knowledge via Knowledge Review

## Reproducibility Summary

**Scope of Reproducibility**

This effort aims to reproduce the results of experiments and analyze the robustness of the review framework for knowledge distillation introduced by Chen et al.[1]. We consistently verify the improvement in test accuracy across student models as reported and study the effectiveness of the novel modules introduced by the authors by conducting ablation studies and new experiments.

**Methodology**

We start the reproduction effort by using the code open-sourced by the authors. We reproduce Tables 1 and 2 from [1] using the same. As we proceed further, we refactor and re-implement the code for a specific architecture (ResNets) and refer to the authors' code for specific implementation details (further discussed in Section 3.2). We implement the ablation studies mentioned in the original paper and design experiments to verify the claims made by the authors. We release our code as open-source.

**Results**

We reproduce the review mechanism on the CIFAR-100 dataset within 0.8% of the reported values. The claim to achieve SOTA performance on the image classification task is verified consistently with different student models. The ablation studies help us understand the significance of novel modules proposed by the authors. The experiments conducted on the framework's components further strengthen the claims made and help get further insights.

**What was easy**

The authors open-sourced the code for the paper. This made it easy to verify many results reported in the paper (specifically Tables 1 and 2 in [1]). The framework of the review mechanism was well described mathematically in the paper, which made its implementation easier. The writing was simple and the diagrams used were self-explanatory, which aided our conceptual understanding of the paper.

**What was difficult**

While the framework of the review mechanism was well described, further specifications of the architectural components, ABF (residual output and ABF output as mentioned in Section 4.2.3) and HCL (number, sizes, and weights of levels as mentioned in Section 4.2.2) could have been provided. These details would make it easier to translate the architecture into code. The most challenging part remained the lack of resources and time to run our experiments. Each run took around 4-5 hours, making it difficult for us to report results averaged over multiple runs.

**Communication with original authors**

During the course of this study, we tried to contact the original authors more than once through e-mail. Unfortunately we were unable to get any response from them.

---

Submitted to ML Reproducibility Challenge 2021. Do not distribute.

# 1 Introduction

Deployment of deep learning models on devices with limited computing capabilities is an active area of research. Knowledge distillation is a popular model compression technique in which 'knowledge' is 'distilled' from an extensive network (called the 'teacher' network) to a smaller network (called the 'student' network), allowing the student to learn better feature representations.

Previous works in knowledge distillation only studied connections paths between the same levels of the student and the teacher, and cross-level connection paths had not been considered. To this end, the CVPR '21 paper *'Distilling Knowledge via Knowledge Review'* [1] presents a new residual learning framework to train a single student layer using multiple teacher layers. This mechanism is analogous to the human learning process, in which a growing child understands and remembers past knowledge as experience. The authors also designed a novel fusion module to condense feature maps across levels and a loss function to compare feature information stored across different levels to improve performance (test accuracy). From here on, we shall refer to the method presented in the original paper as Review KD.

# 2 Scope of reproducibility

Under this reproducibility effort, we focus on reproducing the classification track of the paper. The claims from the original paper, which we attempt to verify, are listed as follows:

1. **Improved classification accuracy for students**: The authors' central claim is that their framework 'achieve[s] state-of-the-art performance on many compact models in multiple computer vision tasks' at a negligible computation overhead. We use a higher test accuracy as a metric for better performance.

2. **Variation of performance across students and teachers**: We study the impact of varying the teacher and student architectures in two scenarios: when the teacher and student belong to the same family and when the teacher and student belong to the different families of networks.

3. **Effect of using HCL instead of $\mathcal{L}_2$ distance**: The authors design a novel loss function which they call the hierarchical context loss (HCL), to compare feature information stored across different levels. They claim that 'the trivial global $\mathcal{L}_2$ distance is not powerful enough to transfer compound levels' information'. We replace HCL with $\mathcal{L}_2$ distance and study its effect on classification accuracy. The architectural details of HCL have also been studied.

4. **Effect of using ABF**: The authors present an attention-based fusion module (ABF), which they argue helps 'obtain a compact framework' and reduce computation. The naïve review mechanism, they claim, 'is straightforward but costly.' We study the impact of attention maps in the ABF module.

# 3 Methodology

We attempt to verify the claims listed in Section 2 using the CIFAR-100 dataset. We use the code made publicly available by the authors on GitHub for validating some of the results reported made in the paper. For conducting ablation studies, we have refactored the authors' code and re-implemented some modules, referring to the architectural descriptions provided in the paper. We also search for the hyperparameters directly related to the distillation mechanism and study the components of the review framework in detail.

## 3.1 Algorithm and Model Description

The review mechanism uses multiple levels of the teacher features to guide a single feature of the student network. All previous (shallower) teacher feature maps guide a given intermediate student feature as shown in Figure 1. The authors design residual learning framework to enable this flow of information. The student feature maps at various levels, that need to be trained by a particular teacher feature, are all fused into a single feature map whose shape is compatible with that of the teacher's. This is shown in Figure 2.

The authors further optimize this mechanism by following a recursive approach to fuse the student features before being trained by the teacher. This approach is shown in Figure 3.

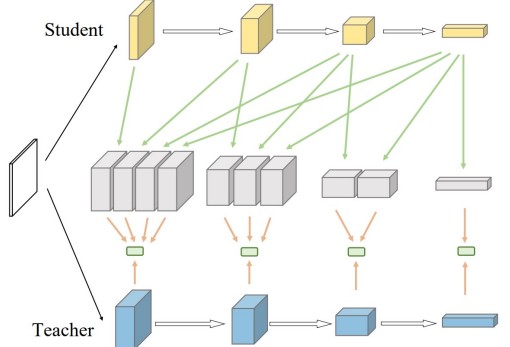

Figure 1: Basic review mechanism (from [1])

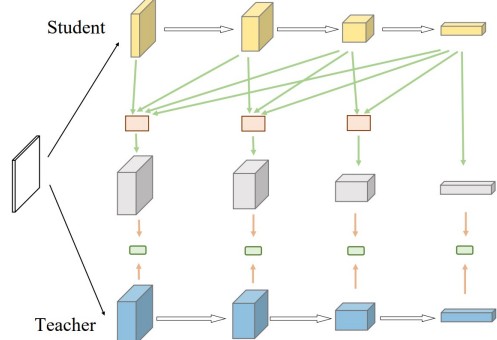

Figure 2: After fusing student features (from [1])



Figure 3: Final review mechanism for knowledge distillation (from [1])

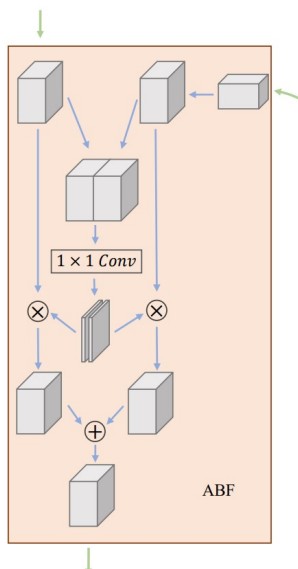

Figure 4: ABF architecture (from [1])

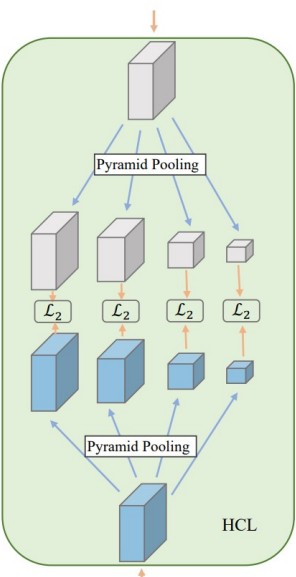

Figure 5: HCL architecture (from [1])

The module used to fuse the current student feature and the previous fused features is the Attention Based Fusion (ABF) module. It involves resizing higher-level features to the same shape as the lower-level features through $1 \times 1$ convolutions and interpolation. The features are then separately concatenated and reduced to two attention maps. The features are multiplied with these attention maps and added to obtain the ABF output.

The distance between the output from the ABF and the teacher feature at that stage is then calculated using the Hierarchical Context Loss (HCL). The features are pooled down to various levels, separating the knowledge at different stages. Between each of these separated levels, the $\mathcal{L}_2$ distance is taken. These distances are then weighted and added together to obtain the final loss at that stage of the student network.

The architectures of ABF and HCL have been shown in Figures 4 and 5.

As students and teachers: we use ResNet20, ResNet32, ResNet56, and ResNet110 (for the CIFAR-100 dataset), which contains three groups of basic blocks with channels of 16, 32, and 64. We also use ResNet8×4 and ResNet32×4, which are ResNets with ×4 channels, 64, 128, and 256, respectively. Wide ResNets are also used and are denoted by WRN-D-R, where D is the depth and R is the width factor. For ShuffleNetV1 and ShuffleNetV2, we use architecture same as that in [2].

## 3.2 Dataset and Training Descriptions

We run our experiments on the CIFAR-100 dataset [3]. It contains 60000 training samples and 10000 test samples of $32 \times 32$. We do not use a validation set, following the authors. The training pipeline uses normalization, random flips, and random crops as data augmentations.

We train all the students and teachers in our experiments for 240 epochs. SGD, along with a Nesterov momentum value of 0.9 is used as the optimizer, along with a weight decay of $5 \times 10^{-4}$. We set the batch size to 128 and initialize the learning rate of 0.1, decaying it to 10 % of its previous value at epochs 150, 180, and 210 for all the experiments and reproductions. This setting matches that used by the authors. For each of our experiments, we report the value of classification accuracy averaged over three runs.

## 3.3 Hyperparameters

We study the hyperparameters involved in the algorithm proposed, with a focus on hyperparameters directly related to the distillation mechanism.

- **Review KD loss weight**: This refers to the weight ($\lambda$) given to the Review KD loss relative to the base loss (Cross-Entropy Loss) in the equation for total loss, that is $\mathcal{L} = \mathcal{L}_{\mathcal{CE}} + \lambda \mathcal{L}_{\mathcal{MKDR}}$. We use a manual approach to search for the best value of Review KD loss weight since each experiment took a significant amount of training time ( $> 3.5$ hours on a single NVIDIA Tesla V100 GPU). The experimental setup and the results obtained have been summarized and visualized in Section 4.2.1.

- **HCL levels**: The hierarchical context loss uses spatial pyramid pooling to transfer the knowledge of various levels separately. The authors use fixed numbers and sizes of feature maps in the levels in their implementation of the HCL. We use this opportunity to study further on how these levels impact the transfer of knowledge. We search over a small hyperparameter space but cover the edge cases to understand the general trend instead of finding the best set of levels. Further details, experimental setup, and results obtained have been summarized and discussed in Section 4.2.2.

- **HCL Levels' Weights**: After extracting the knowledge from different levels using Spatial Pyramid Pooling, the $\mathcal{L}_2$ distance between corresponding layers of teacher and student is the metric used to measure the loss. The authors associate each of these losses obtained (within a single HCL) with a weight (not mentioned in the paper, implemented in code). They use a fixed set of weights to calculate the value of HCL. We study the impact of these weights, keeping the number and sizes of levels fixed. Further details, experimental setup, and results obtained have been summarized and discussed in Section 4.2.2

## 3.4 Experimental setup and code

We use the code open-sourced by the authors to reproduce the results in Tables 1 and 2. Further, we refactored the code for a single architecture (ResNets) to perform our experiments and ablation studies, and for ease of understanding to a new reader. We verified the refactoring by comparing it against the results (test accuracy) obtained using the authors' open-sourced code.

We implement the ablations studied in Section 4.1.3 and the results beyond the original paper in Section 4.2 in PyTorch. Our code can be accessed here.

### 3.5 Computational requirements

For conducting the reproductions and experiments during this study, we use a single NVIDIA Tesla V100 GPU with 32 GB memory provided by our institution. With the description provided in Section 3.2, the number of GPU hours required for training depends mainly on the student used. While training ResNet20/32 and WRN16-1 requires close to 3.7 GPU hours, training ShuffleNets requires around 5-6 GPU hours on the CIFAR-100 dataset.

## 4 Results

### 4.1 Results reproducing original paper

In this section, we reproduce results and tables associated with the CIFAR-100 dataset. We could not use the larger ImageNet dataset [4] due to computing limitations. We cover the student-teacher combinations mentioned in the original paper.

#### 4.1.1 Classification results when student and teacher have architectures of the same style

We first reproduce the results involving the teacher and the student belonging to the same architecture class. We summarise the results obtained in Table 1. The training details are the same as mentioned in Section 3.2. The Review KD loss weight corresponding to each result has been mentioned in the table. The weights used have been referred from the code open-sourced by the authors. This table corresponds to Table 1 reported in the original paper.

We compare the results against test accuracies obtained from previous knowledge distillation methods, including but not limited to variational information distillation [5], probabilistic knowledge transfer [6], and contrastive representation distillation [2]. We observed that our results support the original paper's results.

| Student | ResNet20 | ResNet32 | ReNet8x4 | WRN16-2 | WRN40-1 |
|---|---|---|---|---|---|
| **Test Accuracy** | 69.06 | 71.14 | 72.50 | 73.26 | 71.98 |
| **Teacher** | ResNet56 | ResNet110 | ReNet32x4 | WRN40-2 | WRN40-2 |
| **Test Accuracy** | 73.04 | 74.38 | 79.14 | 76.66 | 76.66 |
| **KD** [7] | 70.66 | 73.08 | 73.33 | 74.92 | 73.54 |
| **FitNet** [8] | 69.21 | 71.06 | 73.50 | 73.58 | 72.24 |
| **PKT** [6] | 70.34 | 72.61 | 73.64 | 74.54 | 73.54 |
| **RKD** [9] | 69.61 | 71.82 | 71.90 | 73.35 | 72.22 |
| **CRD** [2] | 71.16 | 73.48 | 75.51 | 75.48 | 74.14 |
| **AT** [10] | 70.55 | 72.31 | 73.44 | 74.08 | 72.77 |
| **VID** [5] | 70.38 | 72.61 | 73.09 | 74.11 | 73.30 |
| **OFD** [11] | 70.98 | 73.23 | 74.95 | 75.24 | 74.33 |
| **Review KD (Paper)** [1] | 71.89 | 73.89 | 75.63 | 76.12 | 75.09 |
| **Review KD (Ours)** | **71.79** | **73.71** | **76.02** | **76.27** | **75.21** |
| **Review KD Loss Weight** | 0.7 | 1.0 | 5.0 | 5.0 | 5.0 |

Table 1: **Test accuracies** on CIFAR-100 with the teacher and the student having architectures of the same style averaged over three runs

#### 4.1.2 Classification results when student and teacher have architectures of different styles

We then reproduce results that involve distillation when the student's architecture style does not match the teacher's (for instance, a ResNet as a teacher and a ShuffleNet as the student). The training details are the same as mentioned in Section 3.2. The resulting performance of the models and the Review KD loss weight corresponding to each result has been mentioned in the Table 2. The weights used have been referred from the code open-sourced by the authors. This table corresponds to Table 2 reported in the original paper.

| Student | ShuffleNetV1 | ShuffleNetV1 | ShuffleNetV2 |
|---|---|---|---|
| **Test Accuracy** | 70.50 | 70.50 | 71.82 |
| **Teacher** | ResNet32x4 | WRN40-2 | ReNet32x4 |
| **Test Accuracy** | 79.14 | 76.66 | 79.14 |
| **KD** [7] | 74.07 | 74.83 | 73.33 |
| **FitNet** [8] | 73.59 | 73.73 | 73.54 |
| **PKT** [6] | 74.10 | 72.89 | 74.69 |
| **RKD** [9] | 72.28 | 72.21 | 73.21 |
| **CRD** [2] | 75.11 | 76.05 | 75.65 |
| **AT** [10] | 71.73 | 73.32 | 72.73 |
| **VID** [5] | 73.38 | 73.61 | 73.40 |
| **OFD** [11] | 75.98 | 75.85 | 76.82 |
| **Review KD (Paper)** [1] | 77.45 | 77.14 | 77.78 |
| **Review KD (Ours)** | **76.94** | **77.44** | **77.86** |
| **Review KD Loss Weight** | 5.0 | 5.0 | 8.0 |

Table 2: **Test accuracies** on CIFAR-100 with the teacher and the student having architectures of the different style averaged over three runs

These results validate the effectiveness of the distillation algorithm and its ability to transfer knowledge across different architectures. Our results are also consistent with those reported in the original paper.

### 4.1.3 Adding architectural components one by one

After reproducing the test accuracies of various student models for the classification task, we perform ablation studies on the core architecture by adding the components of the review mechanism one by one. This helps us understand the contribution of each component and verify the claims made by the authors. We use a different student-teacher pair as that used by the authors. We aim to reproduce the trend presented in Table 7 of the original paper to verify the generality of the claims.

We train the ResNet20 architecture as the student with ResNet56 as the teacher on the CIFAR-100 dataset. The training details are the same as mentioned in Section 3.2. We also set the Review KD loss weight to 0.6.

The results have been tabulated in Table 3. The baseline is trained with $\mathcal{L}_2$ distance between the same level's features of the student and the teacher. When only the review mechanism is used, the architecture is same as Figure 1. From the third row onwards, when ABF is not used, it is replaced by a fusion module without attention maps, and when HCL is not used, it is replaced by $\mathcal{L}_2$ distance.

The results continuously improve as the the components of the framework are added. Hence, they match the trend obtained by the authors.

| Review Mechanism | Residual Learning Framework | Attention Based Fusion | Hierarchical Context Loss | Test Accuracy |
|---|---|---|---|---|
| | | | | 69.50 |
| ✓ | | | | 69.58 |
| ✓ | ✓ | | | 69.92 |
| ✓ | ✓ | ✓ | | 71.28 |
| ✓ | ✓ | | ✓ | 71.51 |
| ✓ | ✓ | ✓ | ✓ | 71.79 |

Table 3: Adding architectural components one by one

### 4.2 Results beyond original paper

After reproducing the results from the paper, we performed some hyperparameter searches and ablation studies on the components that make up the framework. We try to provide reasoning for the values of hyperparameters set by the authors and the architectural details implemented by them.

#### 4.2.1 Review KD Loss Weight Search

Training ResNet20 as the student with ResNet56 as the teacher, we explore the effect of weight ($\lambda$) given to the Review KD loss relative to the Cross-Entropy loss. The training details are the same as mentioned in 3.2. The results have been visualized in Figure 6.

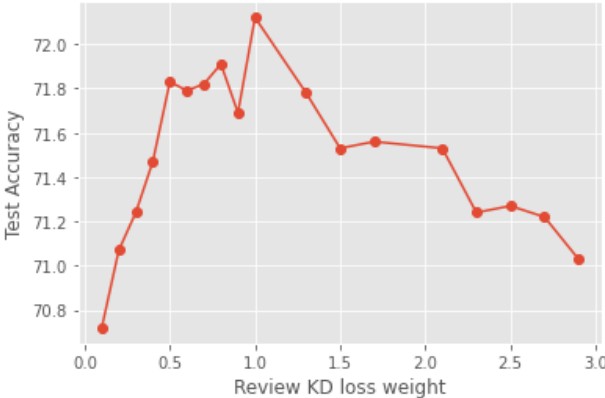

Figure 6: Review KD loss weight vs Test Accuracy for ResNet56 teaching ResNet20 via the Review mechanism

For this student-teacher pair, we find that the best accuracy is obtained when the value of $\lambda$ is in the range between $0.5 - 1.0$. This matches with the value used by the authors while reporting their results. From our results reported in Tables 1 and 2, we note that the optimal value of $\lambda$ depends on the student-teacher combination taken.

#### 4.2.2 Studying HCL in detail

The hierarchical context loss (HCL) is presented as an approach to calculate the distance between two feature maps storing knowledge worth many layers. The loss function splits this knowledge into various levels and finds the $\mathcal{L}_2$ distance separately for each level split, and distills information into the student in different abstract levels.

We train the ResNet20 architecture as the student and ResNet56 as the teacher for the experiments below on the CIFAR-100 dataset. The training details are the same as mentioned in Section 3.2. We also use a Review KD loss weight of 0.6.

**Performance compared to naïve $\mathcal{L}_2$ distance:** Since HCL is introduced as an alternative to $\mathcal{L}_2$ distance, we compare the two on the basis of classification performance. Here, we use the HCL loss used by the authors for other results, i.e., we split the features obtained as output from the ABF into four levels. The first level keeps the ABF output unchanged. The second level is obtained after (max) pooling the ABF output down to $[n, c, 4, 4]$ sized feature maps where $n, c$ is the batch size and the number of channels, respectively. Similarly, the third and fourth levels are $2 \times 2$ and $1 \times 1$ feature maps. The relative weightage given to the $\mathcal{L}_2$ distances obtained after that is $1, 0.5, 0.25, 0.125$, corresponding to each level. On the other hand, we use the $\mathcal{L}_2$ distance between the ABF output and the corresponding teacher features. We obtain a drop of $(0.6 \pm 0.1)\%$ in test accuracy with the above setup. This result agrees with the authors' claim that 'the trivial global $\mathcal{L}_2$ distance is not powerful enough to transfer compound levels' information'.

**Impact of levels in HCL:** To continue our study on the effectiveness of HCL, we decided to vary the number and sizes of feature maps in its levels. We were looking for a trend in what sizes of feature maps in their levels result in the best test accuracy. The results obtained have been summarized in Table 4. An entry, for example, $[h, 4, 2, 1]$ denotes that shapes of feature maps in the levels are $[n, c, h, h], [n, c, 4, 4], [n, c, 2, 2], [n, c, 1, 1]$ where $[n, c, h, h]$ is the shape of the output obtained from ABF. It can be observed that larger-sized feature maps have more impact in transferring

knowledge effectively. In this process, we also obtain test accuracy for this student-teacher combination higher than what is reported in the paper.

| HCL levels | $[4, 1]$ | $[h, 2, 1]$ | $[h, 4, 2, 1]$ | $[h, \lfloor h/2 \rfloor, \lfloor h/4 \rfloor]$ | $[h, h-1, h-2, h-3]$ |
|---|---|---|---|---|---|
| Corresponding Weights | $[1, 0.5]$ | $[1, 0.5, 0.25]$ | $[1, 0.5, 0.25, 0.125]$ | $[1, 0.5, 0.25]$ | $[1, 0.5, 0.25, 0.125]$ |
| Test Accuracy | 71.62 | 71.62 | 71.79 | 71.77 | **71.91** |

Table 4: Impact of levels in HCL

**Impact of weights assigned to levels in HCL:** We then verify our claim made in the previous experiment. While keeping the levels the same as that used by the authors for other experiments, we change the relative weightage of the $\mathcal{L}_2$ distances obtained from each of these levels. The results obtained have been summarized in Table 5. The results of these experiments verify our claim that larger-sized feature maps have more impact in transferring knowledge effectively.

| HCL levels | $[h, 4, 2, 1]$ | $[h, 4, 2, 1]$ | $[h, 4, 2, 1]$ |
|---|---|---|---|
| Corresponding Weights | $[0.125, 0.25, 0.5, 1]$ | $[1, 1, 1, 1]$ | $[1, 0.5, 0.25, 0.125]$ |
| Test Accuracy | 71.46 | 71.75 | **71.79** |

Table 5: Impact of weights assigned to levels in HCL

**Training solely with HCL, no cross-entropy loss:** We explore the limits by entirely omitting the cross-entropy loss and training solely based on HCL. The results obtained were abysmal, being just slightly above the level of random guessing ($\sim 1.5$ %). This result came as a surprise to us, as we expected the loss function to be capable of distilling knowledge independently. We believe this could be explained if we consider the role of HCL as that of a fine-tuner, tuning the weights slightly to achieve performance gain.

### 4.2.3 ABF in detail

The ABF module uses attention maps to aggregate the feature maps dynamically. This allows diverse information from different stages of the network to be fuses reasonably. For the experiments below, we train the ResNet20 architecture as the student and ResNet56 as the teacher for the experiments below on the CIFAR-100 dataset. The training details are the same as mentioned in Section 3.2. We also use a Review KD loss weight of 0.6.

**Fusing without attention maps:** The authors claim that 'the low- and high-level features may focus on different partitions' and 'the attention maps can aggregate them [features] more reasonably'. We attempt to verify this claim by removing the attention maps from the architecture. We then perform fusion using $1 \times 1$ convolutions and interpolation. Once the features are brought into compatible shapes by these operations, they are added directly without multiplying with attention maps. The result is shown in Table 6. We obtain a reasonable drop in accuracy, verifying the effectiveness of attention maps in the architecture.

**With residual output same as ABF output:** Figure 3 shows that the output of ABF module is passed into the next ABF as an input. However, the implementation of ABF module by the authors varies slightly in this regard. The output of the ABF module differs from what is passed into the next ABF as input (residual output). The residual output has a fixed number of pre-defined channels, and the ABF output is generated from the residual output through a $1 \times 1$ convolutional layer. We study the impact of this change in the architecture. The result is shown in Table 6. The result (test accuracy) decreases slightly when the architecture is followed strictly as described in the paper (Figure 3).

| Change in ABF | No change | Fusing without attention maps | With residual output same as ABF output |
|---|---|---|---|
| Test Accuracy | 71.79 | 71.51 | 71.62 |

Table 6: Impact of changes in the ABF architecture

# 5 Discussion

Our aim was to test the basic theoretical and experimental premises of the original paper. Given our computational limitations, we were only able to reproduce the results associated with the CIFAR-100 dataset in this reproduction. The results obtained have supported those reported in the paper. While we could not verify the generality of the approach, we have performed hyperparameter searches and have studied the components in detail. The HCL experiments in Section 4.2.2 and the ABF experiments in Section 4.2.3 further strengthen the claims mentioned in the paper. The experiment where we completely omit the cross-entropy loss also helps gain insights on the working of HCL, and provides a new dimension for further research.

We would like to thank our institution for providing us access to the GPU.

## 5.1 What was easy

The authors open-sourced the code for the paper. This made it easy for us to verify many results reported in the paper (specifically Tables 1 and 2 in [1]). The framework of the review mechanism was well described mathematically in the paper, which made its implementation easier. The writing was simple and diagrams used were self-explanatory, which aided our conceptual understanding of the paper.

## 5.2 What was difficult

While the framework of the review mechanism was well described, further specifications of the architectural components, ABF (residual output and ABF output as mentioned in Section 4.2.3) and HCL (number, sizes, and weights of levels as mentioned in Section 4.2.2) could have been provided. These details would have made it easier to translate the architecture into code. The most challenging part remained the lack of resources and time to run our experiments. As stated earlier, each run took around 4-5 hours, making it difficult for us to report results averaged over multiple runs. In the code open-sourced by the authors, the shapes of output features by the models were hand built in order to make it compatible with the teacher. This made it difficult to use different combinations of teacher and students.

## 5.3 Communication with original authors

During the course of this reproducibility effort, we tried to contact the original authors more than once through e-mail. Unfortunately, we were unable to get any response from them.

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
