# OpenReview forum: "[Re] Distilling Knowledge via Knowledge Review"
_ML_Reproducibility_Challenge/2021/Fall — Reject_

### Official Review · Reviewer_72SH · 2022-02-24
**Great reproducibility effort**

**Rating:** 7
**Confidence:** 3

**Review:**

This report reproduces the results of Distilling Knowledge via Knowledge Review (Chen et al. 2021) on CIFAR-100 dataset, using the code provided by the authors. The report was well structured and the scope of reproducibility was very well established. The report goes beyond the original paper in terms of hyper parameter search and ablation study of the component of ABF & HCL. Over all the report was able to reproduce and validate the contributions of the original paper.

Strengths
* The replication effort is sound and principled.
* Authors provide enough details and code for their replication efforts.
* As far as I could evaluate, authors adhere to the submission template.
* **BONUS** The detailed ablation study of components in HCL and ABF was well thought.

Major concern
* The reproduced report does not have confidence interval. Authors mention that because of the lack of resources and time they were unable to run multiple runs. However, without that it is very difficult to interpret any of the results. Especially given that the difference in accuracies is < 0.3 in many cases.

Minor concerns/Questions
* It is not clear if the KD, FitNet, PKT, RKD etc. baselines in Table 1 & 2 were replicated by the authors or reported directly from the Chen et al.

---

### Official Review · Reviewer_uz3t · 2022-03-04
**Reproducibility Seems to be done right**

**Rating:** 7
**Confidence:** 5

**Review:**

The report seems to be done right, but
1. the authors could not be contacted and
2. the researchers had difficulties with the specifications of the architectural components, and
3. had difficulties in running the code.

The original paper has its issues at the model level, but these have not been mentioned by the researchers.

---

### Official Review · Reviewer_NnH5 · 2022-03-05
**Moderate reproduction**

**Rating:** 6
**Confidence:** 3

**Review:**

This paper reproduces existing CVPR paper called "Distilling Knowledge via Knowledge Review". Specifically, it first reproduce the results using the official open-sourced code, and then re-implement the code by themselves. In addtion, ablation studies are reproduced and new experiments are designed to evaluate its effectiveness. within limited time and resources, this paper reproduces the original paper covering several scopes, including the classification accuracy improvements, performance variation, effectivenss of HCL and ABF. The reproduced results well match that in the original paper.


Minor:
1. We reproduce Tables 1 and 2 from [1] using the same. It seems like it is a incomplete sentence.

---

### Meta-Review · Area_Chair_yTc5 · 2022-04-09

**Recommendation:** Reject
**Confidence:** 5

**Metareview:**

As noted by the reviewers, the report lacks confidence intervals, which are important to measuring experimental variability and reproducibility.  The paper is not accepted.

---

### Decision · Program_Chairs · 2022-04-11

Reject